# Luminescent Hybrid Material Based on Boron Organic Phosphor and Silica Aerogel Matrix

**DOI:** 10.3390/molecules27165226

**Published:** 2022-08-16

**Authors:** Roman Avetisov, Artem Lebedev, Ekaterina Suslova, Ksenia Kazmina, Kristina Runina, Vlada Kovaleva, Andrew Khomyakov, Artem Barkanov, Marina Zykova, Olga Petrova, Alisa Mukhsinova, Denis Shepel, Artyom Astafiev, Natalia Menshutina, Igor Avetissov

**Affiliations:** 1Department of Chemistry and Technology of Crystals, Mendeleev University of Chemical Technology, Moscow 125047, Russia; 2N.N. Semenov Federal Research Center for Chemical Physics, Russian Academy of Sciences, Moscow 119991, Russia; 3Technoinfo Ltd., Stanmore Business & Innovation Center, Hovard Road, London HA7 1GB, UK

**Keywords:** aerogel, luminescence, boron-based phosphor, hybrid material

## Abstract

A new luminescent hybrid material based on silica aerogel and a boron-containing coordination compound with 8-hydroxyquinoline was created, and its physicochemical and spectral-luminescent characteristics were studied. A simple scheme for the synthesis of a hybrid luminescent material was developed. Simultaneously with the synthesis of the aerogel, the formation of a boron-containing phosphor was carried out using an isopropanol solution of boric acid and 8-hydroxyquinoline. Using in situ luminescent measurements, the mechanisms of the formation of boron-based luminescent complexes in isopropanol and tetrahydrofuran media were established. Both hydrophilic and hydrophobic silica aerogels were tested as matrices for the hybrid material. The formation of a thin layer of a boron-containing coordination luminescent compound on the highly developed surface of the SiO_2_ aerogel made it possible to strongly stabilize the aerogel structure and noticeably increase the thermal stability of the synthesized hybrid material.

## 1. Introduction

It is common knowledge that the most effective luminescence occurs on materials’ surfaces. Therefore, a material with a high specific surface can work most efficiently as a luminescent material if the matrix is transparent for its own radiation. New luminescent materials with unique properties are of great interest for the development of novel devices, for example, lighting panels for buildings instead of common lighting devices. When combining the lighting properties with low thermal conductivity, one can obtain a unique material for an outer thermal insulation for the generation of new building constructions. 

This future material could be made on the base of a hybrid material that includes an aerogel, such as an inorganic matrix and an organic phosphor as a lighting source. Such properties are fully possessed by SiO_2_ aerogel.

Silica aerogel is a solid material with a low density from 0.003 to 0.3 g/cm^3^, a high specific surface area up to 1500 m^2^/g, a mean pore size from 10 to 20 nm [1], an extremely low thermal conductivity of about 0.014 W/mK [2], and a high optical transparency in the visible spectral range of about 90%. Due to the cell-net structure, a silica aerogel is an efficient light-scattering inorganic matrix, and its porosity could provide the encapsulation of a large amount of substances, including efficient photoluminophors. Since luminescence is a surface phenomenon, an aerogel with a high specific surface area is an ideal substrate for phosphor formation and bonding at the molecular level to the aerogel structure through covalent bonds or van der Waal bonds. This is the general concept of an aerogel-based luminescent hybrid material.

Recently, we made a hybrid material, *LightSil* [3], based on a silica aerogel and a tris(8-hydroxyquiniline) aluminum (Alq_3_) phosphor by the intercalation of dissolved Alq_3_ into the gel and further supercritical drying. However, taking into account the market price for Alq_3_ of about USD 20,000–100,000 per kg [4], it is hardly possible to produce this hybrid material for building constructions. In addition, there are currently unsolved problems with *LightSil* durability under environmental conditions such as humidity, sunlight, etc.

Therefore, in the present study, we are trying to find a solution for a cheaper method for manufacturing a luminescent hybrid material based on silica aerogel and an organic boron-containing phosphor.

At first glance, boron-based luminescent coordination compounds seem to be unacceptable for the above hybrid synthesis in spite of their high emitting properties. The synthesis procedures of luminescent boron-containing coordination compounds are very complex and, as a rule, they require an argon atmosphere [5,6,7,8]. However, our experiments for luminescent thin film fabrication [9] demonstrated that we could realize the synthesis of a boron-based luminescent compound (BLC) by a simple reaction of boric acid and 8- hydroxyquinoline, assisted by thermal or microwave activations.

The study of the fabrication process and spectral-luminescent properties of a novel luminescent hybrid (*BoronLightSil*) based on a boron complex with 8-Hq as a phosphor and silica aerogel as an inorganic thermal insulating matrix was a goal of the research.

## 2. Materials and Methods

Tetraethoxysilane (TEOS, >99.5%) and trimethylchlorosilane (TMCS, >98%) were purchased from EKOS-1 Ltd. (Moscow, Russia). Hydrochloric acid (HCl, 99.995 wt% 38%), ammonia (99.999 wt% 25%), isopropanol (IPA), n-hexane, and boric acid (H_3_BO_3_, 99.9999 wt%) were purchased from Rushim Ltd. (Moscow, Russia). Boron oxide wires (99.999 wt%) purchased from OCHV.RU, (Moscow, Russia) was dried by dynamic vacuum annealing at 700 °C for 6 h to reach a residual water content less than 50 ppm (according to an IR spectral analysis). 8-Hydroxyquinoline (8-Hq) was purchased from Aldrich and additionally purified by vacuum sublimation to 99.9999 wt%. All reagents were additionally analyzed by ICP-MS (see Section 2.4.3) to prove their chemical purity and to detect possible impurities that had undesirable effects on luminescent properties.

### 2.1. Synthesis of BLCs and Photoluminescence Measurements in Solutions

We used two solvents to conduct the synthesis of BLC, namely, IPA and tetrahydrofuran (TGF). The dissolution of boric acid (H_3_BO_3_) and 8-Hq in the media was carried out at a fixed temperature (45, 50, or 60 °C) in quartz-glass vessels with a preliminary heated solvent in an ultrasonic bath (1.3/2 TTC 35 kHz, Sapphire Ltd., Moscow, Russia) for 60 min. Then, 0.007 g of H_3_BO_3_ and 0.044 g of 8-Hq were dissolved in 0.5 mL of IPA or TGF for different experiments. Then, as-prepared solutions were transferred into a quartz-glass cuvette placed into a 1 cm thermostabilized cuvette holder of an Ocean Optics spectrophotometer. The temperature was supported with 0.1 C accuracy by a BT 12-2 liquid thermostat (TERMEX Ltd., Moscow, Russia). Photoluminescent (PL) spectra were recorded at 365 nm excitation using a QE65000 spectrophotometer (Ocean Optics, Delray Beach, FL, USA). The PL spectral signal was collected for 20 s. Then, we had an exposure for a certain time at a fixed temperature and repeated the PL spectral recording.

### 2.2. One-Step Heterophase Synthesis of BLCs

A heterophase synthesis of solid BLC was conducted using extra-dried boron oxide wires and 8-Hq as raw components. Crystalline P_2_O_5_ was placed outside of the reaction zone and was used as a water-absorbent agent. The general heterophase reaction could be presented by the formal chemical reaction (1) of tris(8-hydroxyquiniline) boron (Bq_3_) synthesis: (1)
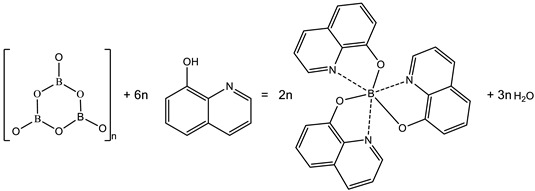


According to [10], the structure of amorphous boron oxide is represented by boron in the triple coordination (flat triangles), which is polymerized into [BO_2_]_3n_ boroxol rings. The proportion of boron polymerized in boroxol rings exceeds 70%. However, one should take into account that the exact B-base product could differ from Bq_3_ because the reaction proceeded on the surface of B_2_O_3_ wires and stopped because of diffusion limitations.

The apparatus scheme for the direct synthesis was described in detail in Ref. [11].

### 2.3. Aerogel Synthesis 

#### 2.3.1. Hydrophilic Boron-Containing Aerogel Synthesis

The process and apparatus for silica gel fabrication was described in detail in Ref. [3]. At the first stage of the fabrication of a boron-containing luminescent aerogel, the gel was placed in a four-fold volume of a solution of H_3_BO_3_ in IPA and left on a shaker for 5–7 h. The procedure for introducing the boron-containing component was repeated three times. The obtained gel was placed in a high-pressure apparatus. Then, the 8-Hq powder preparation was added inside the apparatus in the amount that doubled the stoichiometric amount according to reaction (2).
H_3_BO_3_*^s^* + 3·(8-Hq)^solute^ = Bq_3_*^s^* + 3·H_2_O^solute^(2)

The apparatus was sealed, heated to 40° C, and supplied with CO_2_ up to 120 bar. According to the study of the process of the formation of boron-containing luminescent preparations in IPA solution (see Section 3.1), it was established that we did not need more than 30 min to finish the synthesis reaction in solution. This means that the rate-limiting stage of the BLC formation process inside the gel is the diffusion process. It was found that exposure for 1 h was enough to complete the reaction of BLC synthesis in the gel volume. After that, the supercritical drying of the gels was carried out, and a boron-containing luminescent aerogel (*BoronLightSil*) was obtained.

The total boron content in *BoronLightSil* samples, according to an ICP-MS analysis (Appendix A, Section 2.4.3), turned out to be directly proportional to the boric acid concentration in the IPA solution during the gel impregnation procedure (Figure 1). This indicates that all boron oxide introduced into the aerogel was successfully absorbed in the aerogel porous structure and formed the hybrid material.

#### 2.3.2. Hydrophobic Boron-Containing Aerogel Synthesis

To obtain hydrophobic silica gels, the surfaces of previously obtained gels were subjected to modification. As a result of the modification, hydroxyl groups on the surfaces were replaced by nonpolar groups (alkyl groups) that block the adsorption of water and impart hydrophobic properties to the material. TMCS was used as a hydrophobizing agent. Since TMCS can chemically react with IPA, which had been previously used as a solvent, for the first reaction we changed the solvent to hexane in gel processing by placing the gels in four times the volume of IPA-hexane solutions with different concentrations of hexane (from 30 to 100 wt%) for 3–5 h in each solution. After changing the solvent, the gels were placed in a four times volume of a certain concentration (X wt%) of a TMCS-hexane solution and kept at 60 °C for 24 h in an oven.

After hydrophobization, the samples were washed in IPA to remove unreacted chemicals. The introduction of the boron-containing component and boron complex synthesis via supercritical drying were carried out as described in Section 2.3.1.

Table 1 shows the notation of the hydrophilic and hydrophobic *BoronLightSil* samples and synthesis conditions.

An analysis of the IR spectra showed that the hydrophobization resulted in a sufficient decrease in OH group content (Figure 2).

### 2.4. Measurement Techniques

#### 2.4.1. Aerogel Physical Properties

*BoronLightSil* samples were prepared in the form of cylindrical monoliths. The samples were tested to establish the macroscopic and structural characteristics, i.e., the bulk density (ρ_bulk_) with a standard deviation of 0.005 g/cm^3^; the linear shrinkage (L) with the standard deviation <0.2%; the specific surface area (S_BET_); the total pore volume (V_total_), and the BJH pore volume (V_BJH_). The techniques for characteristic determination were described in detail in Ref. [3]. The specific surface area (S_BET_) was determined by the Brunauer–Emmett–Teller (BET) method. The pore size distribution, mean pore diameter (D), and pore volume (VBJH) were determined by the Barrett–Joyner–Halenda (BJH) method. An AccuPyc 1340 instrument (Micromeritics Instrument Corp., Norcross, GA, USA) was used to measure the skeletal density (ρsk) of *BoronLightSil* by helium pycnometry.

#### 2.4.2. Luminescence Measurements

All spectra, except those for solutions (see Section 3.1), were recorded at room temperature.

A Fluorolog-3 FL3-22 spectrophotometer (Horiba Jobin Yvon, Portland, OR, USA) with double grating excitation and emission monochromators was used for the photoluminescence (PL) measurements of solid samples and aerogels in the wavelength range of 375–725 nm with 0.1 nm steps under the excitation of a Xenon 450W Ushio UXL-450S/O lamp.

The PL spectra and decay kinetics were treated by OriginPro 8 SR4 software (OriginLab Corp., Northampton, MA, USA), taking into account the original HORIBA software for instrumental corrections [12]. Photoluminescence quantum yield was determined using a G8 integration sphere (GMP SA, Renens, Switzerland).

The calculations of the chromaticity coordinates of the luminescence, according to the CIE RGB colorimetric system of HM films, were carried out using the addition curves.

#### 2.4.3. ICP-MS Analysis

To analyze the chemical purity of the initial preparations and synthesized products, we used inductively coupled plasma mass spectrometry with the preliminary transfer of a solid sample to the liquid phase. For dissolving preparations, we used extra pure water (Aqua-Max-Ultra 370 Series, Young Lin Instruments Co. Ltd., Gyeonggi-do, 14042, Korea) with a specific resistance of 18 MΩ·cm, high-purity nitric (HNO_3_) acid (7N7) purified by a Berghoff BSB-939-IR surface distillation system (BERGHOFF GmbH&Co., Wenden, Germany), and high-purity sulfuric acid (H_2_SO_4_) (8N Ultrapur, Sigma Aldrich, Burlington, MA, USA) in a SPEEDWAVE-FOUR microwave decomposition system (Berghof GmbH, Koenigsee, Germany) equipped with DAP-100 PTFE autoclaves (BERGHOFF GmbH&Co.). 

Analytical measurements were carried out on a NexION 300D inductively coupled plasma mass spectrometer (ICP-MS) (PerkinElmer Inc., Boston, MA, USA). We used the TotalQuant method [13] for the determination of 65 chemical elements’ concentrations. The standard solutions (PerkinElmer Inc.) were used for calibration.

#### 2.4.4. TEM Analysis

Micrographs of the samples were obtained using JEM 2100F and JEM 2100M electron transmission microscopes (maximum accelerating electron voltage—200 kV, chromatic and spherical aberration correctors, and X-ray energy-dispersive analyzer). The sample preparation for electron microscopy consisted of the deposition of small BLC strips (thickness less than 100 µm) on copper and carbon grids, without the deposition of additional materials. Moderate sample drift was observed only at maximum magnifications (2 M or more).

## 3. Results and Discussion

### 3.1. Boron-Based Luminescent Complexes

When we studied the solution synthesis of BLC. We found that in 30 min the maxima of the PL wavelengths ($\lambda_{PL}^{max}$) reached constant values, which indicated that the synthesis reaction of BLC came to an end (Figure 3).

The most interesting result was obtained for the analysis of the PL spectra recorded at different reaction temperatures. It turned out that during the reaction two different boron-based luminescent complexes were formed. The complexes differed in the $\lambda_{PL}^{max}$ value. The transition from one temperature to another led to a redistribution of the concentrations of these complexes, as evidenced by the redistribution of the intensities of the PL bands for $\lambda_{PL}^{max}$.

A decomposition of the PL spectra in energy coordinates (Figure 4) made it possible to determine that one complex (type **I**) was characterized by $\lambda_{PL}^{max}$ = 535 nm, while the second complex (type **II**) had $\lambda_{PL}^{max}$ = 427 nm.

It is well-known that 8-Hq complexes with Al, Ga, and In have a number of polymorphic modifications that are characterized by differences in structure, and their $\lambda_{PL}^{max}$ values differ by a maximum of 50 nm [14,15,16]. In our case, the difference in $\lambda_{PL}^{max}$ was almost 100 nm, which can hardly be explained by the formation of different polymorphic modifications. Taking into account the complex polymer formations based on boron oxide groups in the liquid, it can be assumed that at different temperatures different degrees of coordination of the boron complex with 8-Hq arise. It is known that a decrease in the mass of the coordination metal and the amount of ligands for metal complexes with 8-Hq generally leads to a decrease in $\lambda_{PL}^{max}$ [17]. Thus, in the row Alq_3_→Caq_2_→Liq for structurally similar polymorphic modifications, a regular shift of $\lambda_{PL}^{max}$ is seen as 520 nm→465 nm→442 nm, respectively. Thus, it can be assumed that boron in the row B→Al→Ga→In, having the smallest molecular weight, forms complexes characterized by the smallest values of $\lambda_{PL}^{max}$. In this case, the coordination of one ligand leads to the generation of luminescence with the shortest $\lambda_{PL}^{max}$. We supposed that this effect was observed in our experiments.

When using tetrahydrofuran as a solvent, taking into account the much better solubility of both boric acid and 8-Hq in TGF over the entire temperature range under study from 30 to 60 °C, the maximum PL intensity was almost instantly obtained, with a constant maximum of $\lambda_{PL}^{max}$ = 535 ± 2 nm (Figure 5). Taking into account that the boron oxide groups in TGF are less polymerized, it can be assumed that, in this case, complexes similar in structure to Bq_3_ are formed. 

To exclude a solvent intercalation in a BLC structure, which could take place during solution synthesis, we analyzed the BLC preparations obtained by the heterophase reaction (1). Similar to IPA solution synthesis, we obtained two preparations that strongly differed in PL intensity and $\lambda_{PL}^{max}$ (Figure 6). 

Since the entire preparation was practically in the isothermal zone, we assumed that the luminescence color was influenced by the 8-Hq vapor flow and the outgoing water vapor. The preparation located closer to the 8-Hq source showed a turquoise glow (B-Hq-Dir(493)) of PQY 55%, while the preparation located far from the 8-Hq source glowed blue (B-Hq-Dir(459)) at PQY 41%.

An analysis of the PL decay kinetics showed that both preparations were characterized by the presence of two luminescence centers with different lifetimes (Figure 7, Table 2). As follows from the fundamental regularities, the complex with $\lambda_{PL}^{max}$ = 459 nm was characterized by a faster short-lived component (2.16 ns), while the complex with $\lambda_{PL}^{max}$ = 494 nm had a slower short-lived component (6.42 ns). For the long-lived luminescence centers, the lifetime values were close (32.3 ns vs. 29.6 ns).

In the case of boron oxide wires, their surface is smooth enough, with separate pyramids (Figure 8a). BLC could form on a smooth surface and on the sharp edges of wires or pyramids. Thus, we obtained two possible PL centers with different lifetimes.

The probable reaction of BLC synthesis on a smooth surface could be presented by the scheme in Figure 9, resulting in a monomeric coordination compound (B-Hq-Dir(459)).

On the sharp edges of ledges that existed on a smooth wire surface (Figure 8b), in accordance with steric factors, several 8-Hq molecules could interact with boron, resulting in a dimeric coordination compound with a brighter PL (B-Hq-Dir(493)) (Figure 10).

BLC stability was proven by cathodoluminescence (CL) measurements of a B-Hq-Dir(493) preparation. For the first time, we detected CL for BLC based on 8-hydroxyquinoline (Figure 11). The CL was detected in blue and green channels, but the red channel was empty (Appendix A). We observed CL only on the grains. The thin layers of BLC on the smooth surfaces of B_2_O_3_ wires were probably quickly sputtered by the electron beam at 20 kV.

The TEM analysis showed the presence of many different objects in BLC samples obtained by heterophase synthesis, which could be geometrically divided into four types: (1) clusters with a shell (Appendix A); (2) clusters without a shell (Figure 12, Figure 13, Appendix A); (3) nanotubes (Figure 14 and Appendix A); and (4) crystalline planes, including partially twisted ones (Appendix A). It is most likely that nanotubes are formed from twisted planes. This means that we observed nanotubes at different stages of their formation.

Unfortunately, we failed to detect structures containing 8-hydroxyquiniline, whose crystal lattice parameter is about 6–8 Å in coordination compounds [15]. This could be explained by very the small thickness of BLC, about several monolayers. These thin structures could be easily vaporized by the e-beam of TEM, similar to the process we observed when we analyzed tris(8-hydroxyquiniline) aluminum crystals by SEM. We observed the crystal vaporization under e-beam action in real time for several seconds.

The observed crystal lattices can be classified into two types:Lattice parameter 3.2 ± 0.5 Å. Taking into account the limited number of elements in the reaction components (H, C, N, O, and B), we searched for appropriate crystalline structures and came to the conclusion that the only suitable phase with a 3.2 ± 0.5 Å lattice parameter could be C_3_N_4_ [18]. There are several polymorphs of C_3_N_4_ that meet the search lattice parameter. The small clusters of C_3_N_4_ could be formed as a result of 8-Hq decay at high temperatures and crystal lattice compression. This assumption needs further investigation, but it could be very interesting to synthesize C_3_N_4_ in such soft conditions.The lattice parameter of 4.2 ± 0.5 Å could be attributed to crystalline B_2_O_3_ (space group P3_1_21) [19].

The main conclusions after image processing: Single-crystal planes are mainly formed by a lattice with a lattice parameter of 3.2 Å. The fraction of planes with a lattice of 4.2 Å is very small and their size is an order of magnitude smaller than that of the former.The same 3.2 Å lattice forms nanosized clusters with diameters of up to 20 nm of a rather spherical shape (Figure 12).A crystal lattice with a parameter of 4.2 Å also forms clusters, including two types of small irregularly shaped clusters up to 10 nm in size: four neighboring atoms in this lattice form a rhombus with an angle of 60 degrees (Figure 13) and clusters with a shell (dot sizes of 8–12 nm and core sizes of 4–6 nm, Appendix A).Nanotubes are formed only by a crystal lattice with a parameter of 3.2 Å (Figure 14). The dimensions of the tubes are small; the length range is from 20 to 50 nm, with diameters of 4 to 15 nm. We also observed multilayer tubes (up to 16 layers, with 2.4 A between layers).

### 3.2. BoronLightSil

An analysis of the physicochemical properties and structural parameters of silica aerogels and *BoronLightSil* samples (Table 3) showed that the bulk density (**ρ_bulk_**) of silica aerogels was half that of the hydrophilic *BoronLightSil* samples, while the skeletal densities (ρ_sk_) differed only by 15%. Eventually, the total pore volume of the silica aerogel was twice as large as in the *BoronLightSil* samples due to the pores filling with boron oxide and BLCs.

Another situation was observed for hydrophobic *BoronLightSil* samples. The bulk densities for *BoronLightSil* were two times higher than in hydrophobic silica aerogel. The skeletal densities were nearly the same as for the hydrophilic *BoronLightSil* samples and just a bit lower the hydrophobic silica aerogel. The total pore volume was bigger than for the hydrophilic samples for both the silica aerogel and *BoronLightSil* samples.

The PL spectra of hydrophilic *BoronLightSil* samples (Figure 15) could be characterized as simple and attributed to a single chemical compound, contrary to the solution synthesis of BLC (see Section 3.1). The $\lambda_{PL}^{max}$ of the hydrophilic *BoronLightSil* samples was 513 ± 5 nm (Table 4) and was slightly shifted to the red wavelength region on boron concentration increase.

An analysis of the PL spectra of the *BoronLightSil* samples showed that an increase in the total boron concentration in the hydrophilic *BoronLightSil* samples first led to an increase in the PL intensity (Figure 15) and then to a noticeable decrease.

This character of dependence can be explained by the fact that boron oxo groups, having a size of 5–7 Å, are formed at hydrophobic ends and also penetrate into the pores of aerogels and fill them. When interacting with 8-hydroxyquinoline, the exchange reaction (1) results in the formation of organo-element monomeric [6], dimeric [7], or even more complex clusters, which have sizes of about 2–4 nm. These complexes are easily located on hydrophobic ends and inside pores. However, if the pores are heavily filled with boron oxy groups, then the reaction of BLC formation proceeds efficiently at the exit from the pores and does not propagate deep inside due to steric limitations. Thus, boron oxy groups inside pores are not involved in the formation of luminescent complexes and can reduce the luminescence intensity, which we observed experimentally (Figure 15).

In the case of hydrophobic *BoronLightSil* samples, we observed the PL intensity to be one order higher (Figure 16, Table 4). The $\lambda_{PL}^{max}$ values were shorter (494 and 498 nm) compared to the hydrophilic samples, but were very close to those of the direct-synthesized BLC (see Table 2).

This could indicate that the synthesis reactions of BLC in hydrophobic aerogels are very similar to those that occurred in the direct synthesis at high temperatures (Section 2.2).

To understand the nature of BLC formation, we analyzed the PL decay kinetics of *BoronLightSil* samples (Figure 17, for details see Appendix A).

According to an analysis, for all *BoronLightSil* samples the PL decay kinetics were successfully described by three exponents (Table 4, Appendix A), contrary to the solid direct-synthesized BLC samples (Table 2), where we found two luminescent centers. For most of the *BoronLightSill* samples, the short-lived centers had lifetimes of about 2.4 ± 0.1 ns. The exception was the BLS-0.05-HB-5 sample, in which hydrophobization was more complete due to the high concentration of TMCS. The middle-lived centers had lifetimes of about 11 ± 2 ns for all samples, and the long-lived centers for hydrophilic *BoronLightSil* had lifetimes of 168 ± 10 ns, while for hydrophobic *BoronLightSil* they had shorter lifetimes of 74–148 ns.

The QY of hydrophobic *BoronLightSil* was compared with BLC samples, while for hydrophilic *BoronLightSil* the QY was lower at times. We explained the obtained results due to the known poor resistance of 8-hydroxyquinolne-based coordination compounds under humidity exposure [20]. The presence of OH groups in hydrophilic *BoronLightSil* decreased the PL intensity by more than one order of magnitude, and the QY reduced correspondingly.

We assume that the observed results could be explained by the differences in structures of boron oxide wires and silica aerogels. In the case of *BoronLightSil*, its aerogel matrix has a much more complex structure with bridges and pores of different sizes (Figure 8b). We could identify at least three different locations for BLC formation, namely, smooth bridge wires, pore exits, and pore walls. Therefore, the BLC configuration at these sites is different, resulting in different lifetimes. We assumed that the shortest lifetime would be for BLC located on a smooth surface and that the BLC located on the walls of the pores would have the longest lifetime.

The hydrophobization of *BoronLightSil* aerogels made it possible to significantly increase the stability of luminescent characteristics. In the case of a hydrophilic *BoronLightSil* sample, the loss of PL intensity was almost 100% within 6 months of exposure to a natural environment (Figure 18a), while for the hydrophobic *BoronLightSil*, the decrease in the PL intensity after the same 6 months of exposure was only 15% (Figure 18b). It should be notice that a *LightSil* hybrid based on Alq_3_ [3] had a PQY of 45% (measured in this research), but it decayed in several days under natural environmental conditions.

An analysis of the thermostability of *BoronLightSil* was conducted by the step-by-step heating of a sample placed into a thermostabilized quartz-glass cuvette and in situ PL measurements. Ten minutes of exposure was enough to reach a constant PL intensity value at a given temperature. We observed a continuous decrease in PL intensity during a temperature increase (Figure 19). However, up to 90 °C the aerogel structure of the *BoronLightSil* sample was preserved, while the ordinary SiO_2_ aerogel decayed at 80 °C. Taking into consideration the thermal stability of Mq_3_ (M = Al, Ga, and In) compounds [14], we assume that a thin layer of BLC on a highly developed SiO_2_ aerogel surface strongly stabilized the aerogel structure and noticeably increased its thermal stability.

## 4. Conclusions

A new luminescent hybrid material based on silica aerogel and a boron-containing coordination compound with 8-hydroxyquinoline, named *BoronLightSil*, was created. We developed a simple scheme for the synthesis of a hybrid boron-based luminescent material that combined the processing of hydrophobic silica aerogel fabrication by supercritical drying in CO_2_ flow at 120 atm and the simultaneous synthesis of a boron-based phosphor in situ by the chemical reaction of boric acid with 8-hydroxyquinilinol in isopropanol. The study of the chemical reaction in isopropanol and tetrahydrofuran media showed that the nature of the luminescent complexes based on boron with 8-hydroxyquinoline is very complex. At least two basic products could be formed, depending on the reaction conditions.

It was found that the hydrophobization of silica aerogels by a treatment in n-hexane resulted in a *BoronLightSil* material with stable luminescent properties and higher thermostability.

## Figures and Tables

**Figure 1 molecules-27-05226-f001:**
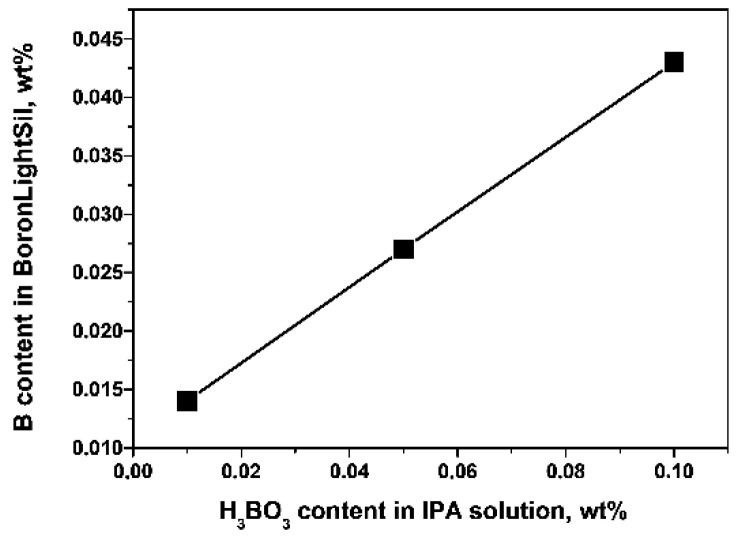
Boron concentration in the synthesized *BoronLightSil* preparations vs. H_3_BO_3_ concentration in IPA solution during the gel impregnation.

**Figure 2 molecules-27-05226-f002:**
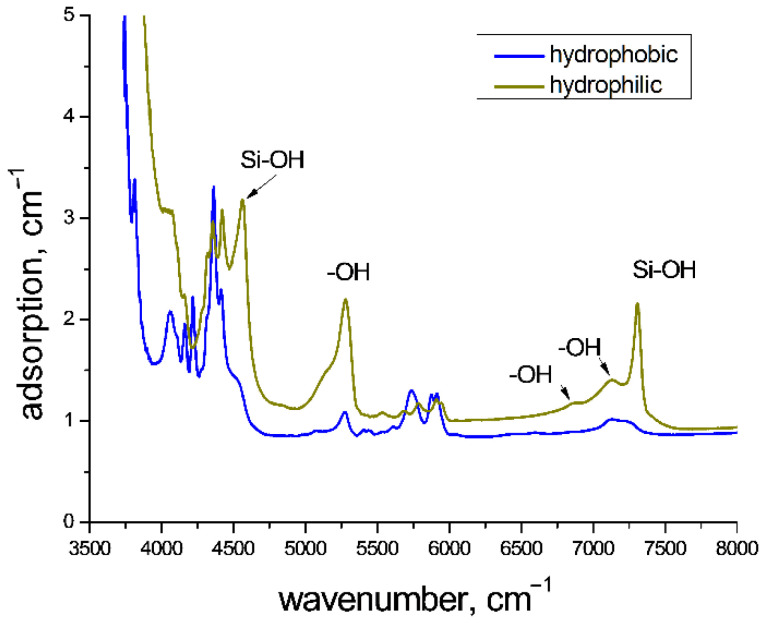
Adsorption spectra of hydrophobic and hydrophilic SiO_2_ aerogels.

**Figure 3 molecules-27-05226-f003:**
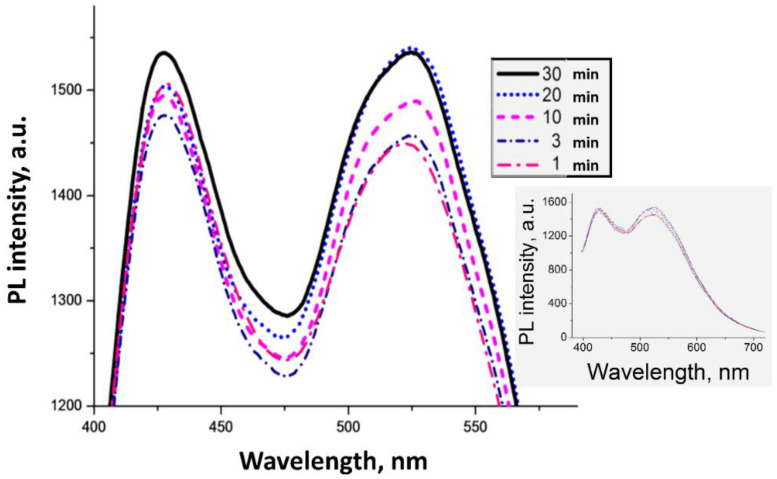
PL spectra of (H_3_BO_3_ + 8-Hq) solution in IPA, depending on the exposure time at 50 °C (λ^ext^ = 365 nm).

**Figure 4 molecules-27-05226-f004:**
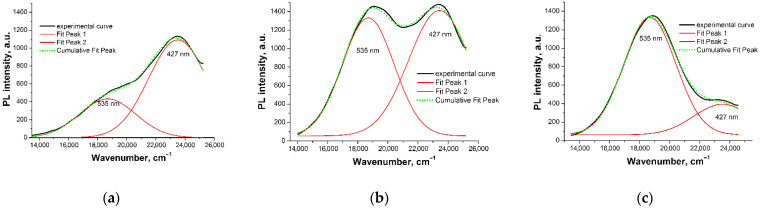
PL spectra of (H_3_BO_3_ + 8-Hq) solution in IPA after 30 min exposure at (**a**) 45 °C, (**b**) 50 °C, and (**c**) 60 °C (λ^ext^ = 365 nm).

**Figure 5 molecules-27-05226-f005:**
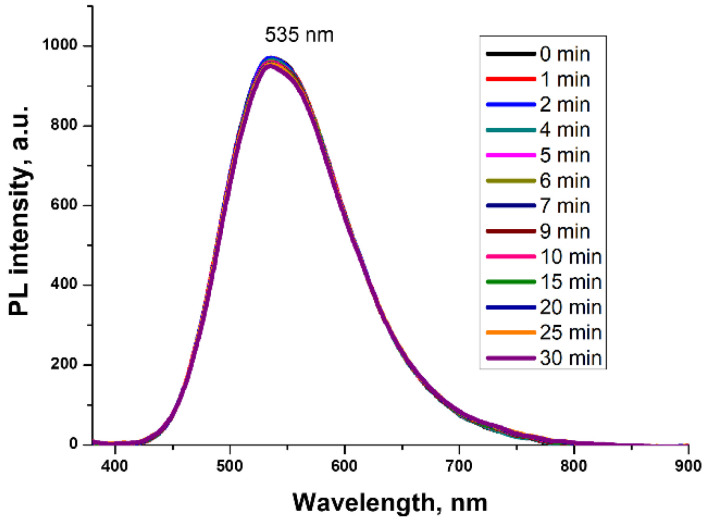
PL spectra of (H_3_BO_3_ + 8-Hq) solution in TGF, depending on the exposure time at 30 °C (λ^ext^ = 365 nm).

**Figure 6 molecules-27-05226-f006:**
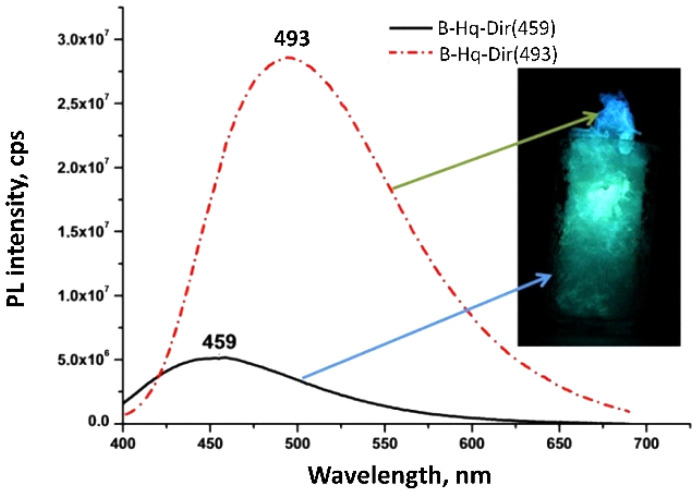
PL spectra of BLC obtained by heterophase synthesis.

**Figure 7 molecules-27-05226-f007:**
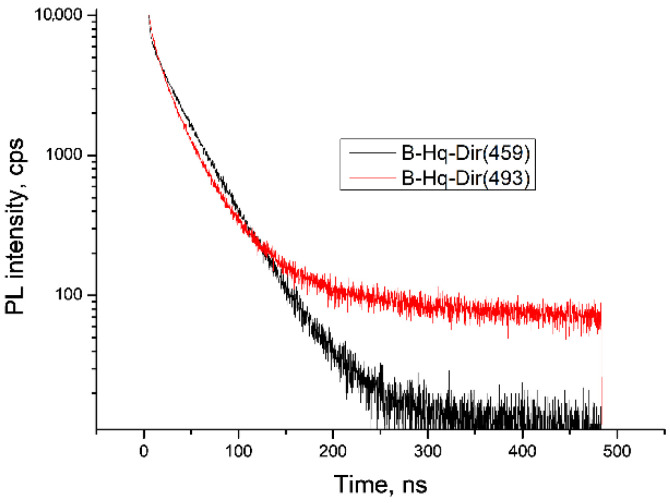
PL decay kinetics of BLC preparations obtained by heterophase synthesis.

**Figure 8 molecules-27-05226-f008:**
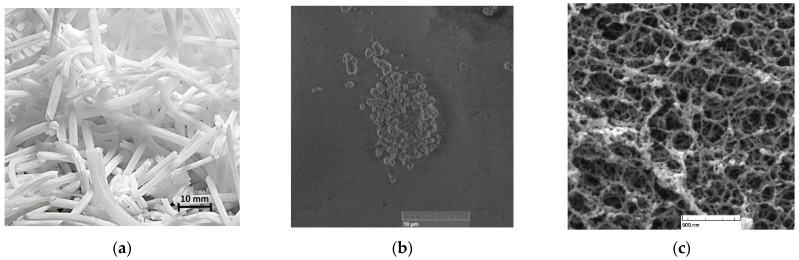
Photo of boron oxide wires (**a**) and SEM image of BLC (**b**) and silica aerogel (**c**).

**Figure 9 molecules-27-05226-f009:**
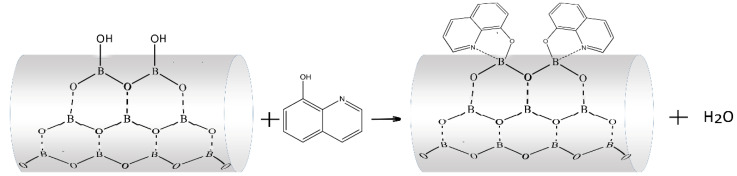
Scheme of the BLC formation of a smooth wire wall of boron oxide.

**Figure 10 molecules-27-05226-f010:**
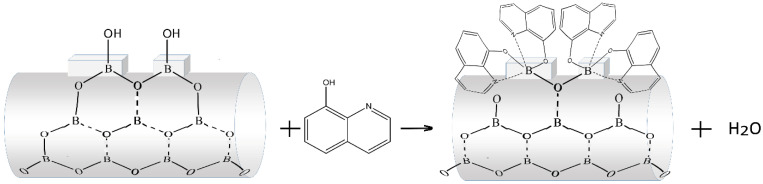
Scheme of the BLC formation on the sharp edges of ledges of boron oxide wires.

**Figure 11 molecules-27-05226-f011:**
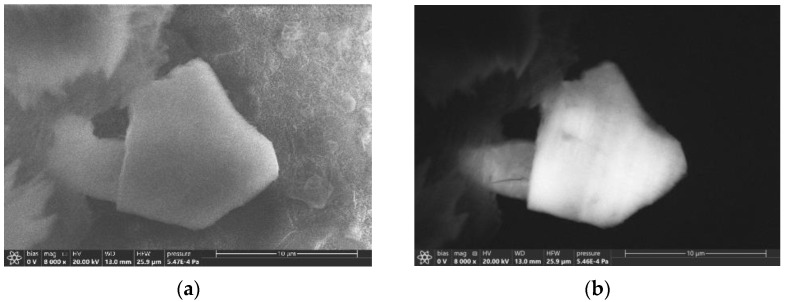
SEM images of a B-Hq-Dir(493) BLC preparation fabricated by direct synthesis at 20 kV accelerating voltage: (**a**) SEM image, (**b**) total cathodoluminescence.

**Figure 12 molecules-27-05226-f012:**
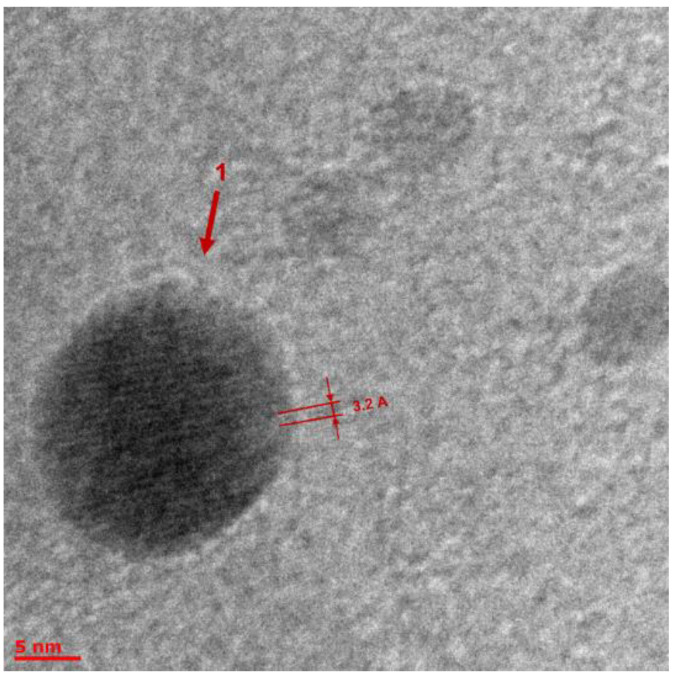
TEM image of Object 1 as a spherical cluster. The lattice parameter is 3.2 Å, and the cluster size is 20 nm.

**Figure 13 molecules-27-05226-f013:**
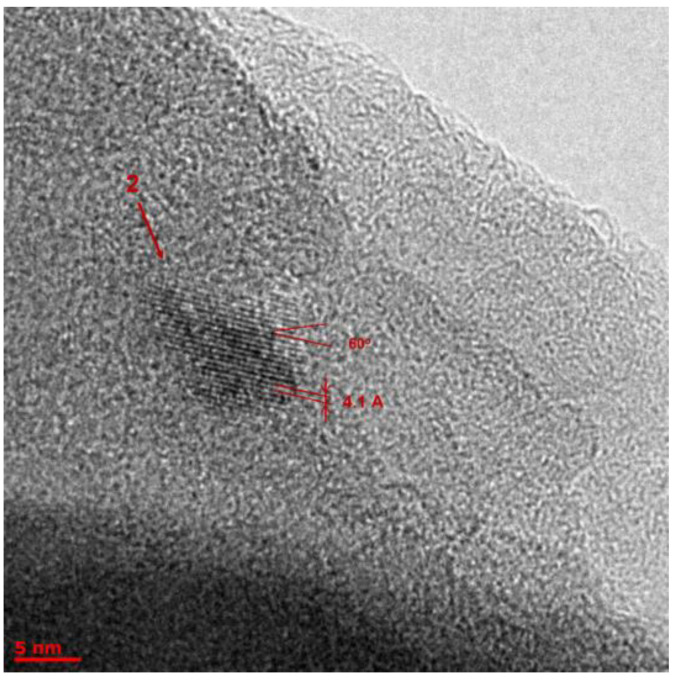
TEM image of Object 2 as a cluster of complex shape. The lattice parameter is 4.2 Å. The angle between three neighboring atoms in the plane is 60 degrees (rhombus).

**Figure 14 molecules-27-05226-f014:**
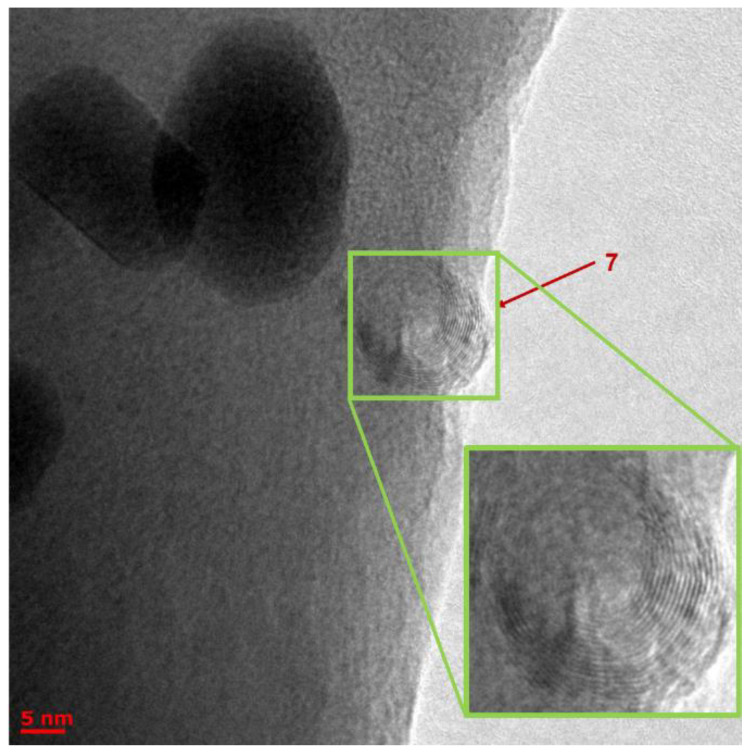
TEM image of Object 7 is a multilayer nanotube with 16 layers. The distance between the layers is 2.4 Å, and the distance between neighboring atoms is about 3.2–3.6 Å. The total diameter of the object is 10 nm.

**Figure 15 molecules-27-05226-f015:**
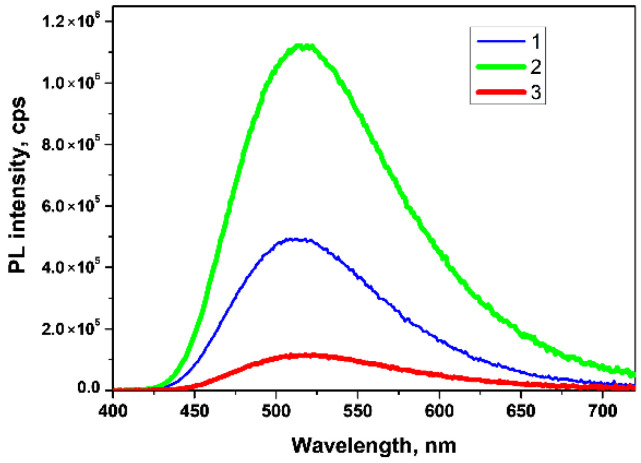
PL spectra of hydrophilic *BoronLightSil* samples with different concentrations of H_3_BO_3_ during gel synthesis: (1) 0.01 wt% for BLS-0.01-HL; (2) 0.05 wt% for BLS-0.05-HL; (3) 0.1 wt% for BLS-0.10-HL (λ^ext^ = 360 nm).

**Figure 16 molecules-27-05226-f016:**
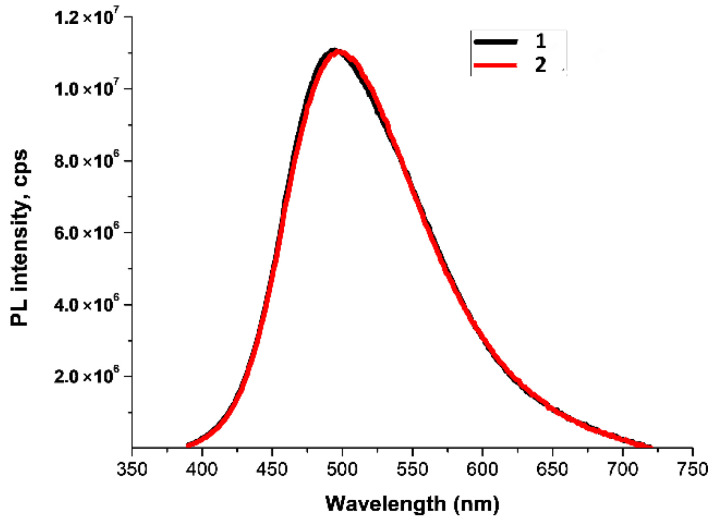
PL spectra of hydrophobic *BoronLightSil* samples with different concentrations of TMCS: (1) 3% for BLS-0.05-HB-3 sample and (2) 5% for BLS-0.05-HB-5 sample.

**Figure 17 molecules-27-05226-f017:**
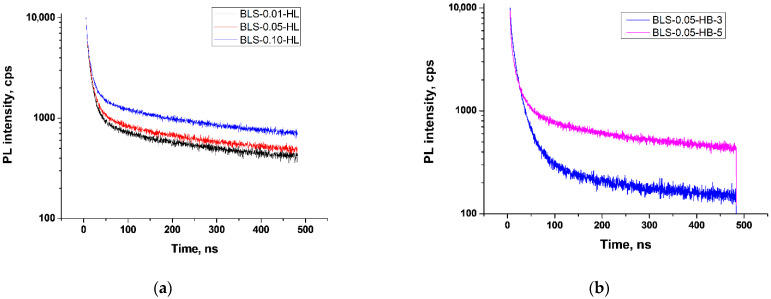
PL decay kinetics of (**a**) hydrophilic and (**b**) hydrophobic *BoronLightSil* samples.

**Figure 18 molecules-27-05226-f018:**
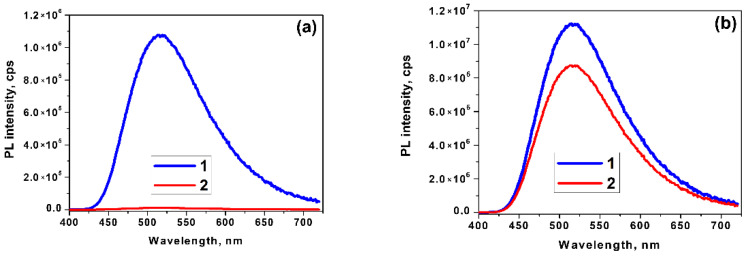
PL spectra of *BoronLightSil* samples synthesized using (**a**) hydrophilic and (**b**) hydrophobic aerogels: 1—samples immediately after synthesis; 2—after exposure to air for 6 months.

**Figure 19 molecules-27-05226-f019:**
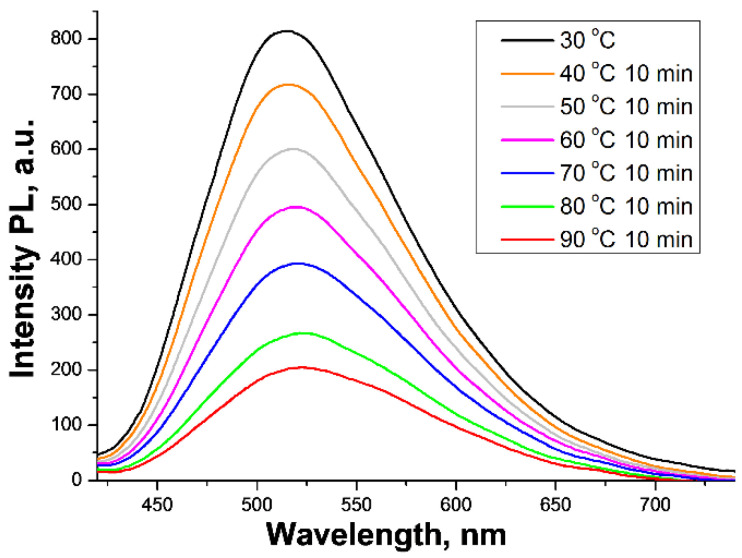
PL spectra of hydrophobic *BoronLightSil* sample during continuous heating with 10 min exposure at different temperatures (λ^ext^ = 365 nm).

**Table 1 molecules-27-05226-t001:** Notation of the obtained *BoronLightSil* samples.

Sample ID	H_3_BO_3_, wt%	TMCS X, wt%	Water Affinity
BLS-0.01-HL	0.01	0	Hydrophilic
BLS-0.05-HL	0.05	0	Hydrophilic
BLS-0.10-HL	0.10	0	Hydrophilic
BLS-0.05-HB-3	0.05	3	Hydrophobic
BLS-0.05-HB-5	0.05	5	Hydrophobic

Note: HL—hydrophilic; HB—hydrophobic.

**Table 2 molecules-27-05226-t002:** PL characteristics of BLC preparations obtained by heterophase synthesis.

Sample ID	PL Peak Characteristics				Lifetime ns		CIE Color Coordinate	
	Area, nm × cps	FWHM, nm	Center, nm	Height, cps	τ1	τ2	X	Y
B-Hq-Dir(459)	0.59 × 10^9^	107	459	0.49 × 10^7^	2.16	32.3	0.2839	0.4840
B-Hq-Dir(493)	1.43 × 10^9^	117	493	2.72 × 10^7^	6.42	29.6	0.3022	0.4860

**Table 3 molecules-27-05226-t003:** Physicochemical properties and structural parameters of silica aerogels and *BoronLightSil* samples.

Sample ID	ρ_bulk_, g/cm^3^	L, %	S_BET_, m^2^/g	D, nm	V_BJH_, cm^3^/g	V_por_, cm^3^/g	ω, %	ρ_sk_, g/cm^3^	ε
HL	0.114	6.2	1087	15.2	4.4	8.3	53.1	2.07	0.94
HB	0.089	3.4	1239	10.3	3.5	10.7	32.6	1.95	0.95
BLS-0.01-HL	0.212	19.5	1021	15.7	3.2	4.1	77.4	1.70	0.88
BLS-0.05 HL	0.226	21.2	1010	13.4	3.1	3.8	80.8	1.72	0.87
BLS-0.10-HL	0.203	18.4	1015	14.2	3.7	4.3	85.1	1.70	0.88
BLS-0.05-HB-3	0.148	4.35	1186	9.8	3.1	6.2	50.3	1.68	0.91
BLS-0.05-HB-5	0.142	2.34	1150	9.1	3.0	6.5	46.4	1.72	0.92

**Table 4 molecules-27-05226-t004:** Photoluminescent characteristics of *BoronLightSil* samples.

Sample ID	PL Peak Characteristics				Lifetime, ns			CIE Color Coordinate		QY, %
	Area, nm × cps	FWHM, nm	Center, nm	Height, cps	τ1	τ2	τ3	X	Y	
BLS-0.01-HL	0.59 × 10^8^	107	509	0.49 × 10^6^	2.3	9.9	158	0.2839	0.4840	24
BLS-0.05-HL	1.43 × 10^8^	117	513	1.12 × 10^6^	2.5	10.5	168	0.3022	0.4860	33
BLS-0.10-HL	0.15 × 10^8^	112	518	0.16 × 10^6^	2.4	10.6	180	0.3140	0.4991	11
BLS-0.05-HB-3	1.39 × 10^9^	115	494	1.11 × 10^7^	2.5	13.0	74	0.2472	0.3939	45
BLS-0.05-HB-5	1.39 × 10^9^	114	498	1.10 × 10^7^	1.3	11.9	148	0.2480	0.3983	44

## Data Availability

Not applicable.

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
