# Peer review of "Luminescent Hybrid Material Based on Boron Organic Phosphor and Silica Aerogel Matrix"

_molecules, 2022, doi:10.3390/molecules27165226_

Round 1

Reviewer 1 Report

Date: 26-07-2022

Ms. Ref. No.:  molecules-1847245

Title: “Luminescent hybrid material based on boron organic phosphor and silica aerogel matrix”.

Recommendation: Minor revision

The authors of the manuscript titled “Luminescent hybrid material based on boron organic phosphor and silica aerogel matrix” fabricated a luminescent hybrid material based on silica aerogel and a boron-containing coordination compound with 8-hydroxyquinoline and their physico-chemical and spectral-luminescent characteristics have been reported. To be published in molecules journal the quality of the manuscript should be significantly improved.

The authors should therefore emphasis on the following points   

1.      The introduction is very poor, which should be improved. The authors stated, “For the first look boron based luminescent coordination compounds seems to be unacceptable for the above hybrid synthesis due to the complex synthesis procedure of boron phosphors [5] in spite of their high emitting properties”. Explain the existing hybrid synthesis procedures and distinguish between the existing and current synthesis process. The quantitative description about emission properties should be discussed.  

2.      The detailed luminescence analysis should be provided. What about the quantum efficiencies of the hybrid materials?

3.      It will be interesting to check the thermal stability of the hybrid materials.

4.      Comment on the stability of the boron-based materials?

5.      Thorough comparison between Alq3 and current material should be provided considering both their synthesis and luminescence properties to stablish the correlation between them.    

Author Response

The authors should therefore emphasis on the following points  

  1. The introduction is very poor, which should be improved. The authors stated, “For the first look boron based luminescent coordination compounds seems to be unacceptable for the above hybrid synthesis due to the complex synthesis procedure of boron phosphors [5] in spite of their high emitting properties”. Explain the existing hybrid synthesis procedures and distinguish between the existing and current synthesis process. The quantitative description about emission properties should be discussed.

Reply

We added the description of problems with standard boron phosphor synthesis, and presented the general description of our approach and possible privileges.

  1. The detailed luminescence analysis should be provided. What about the quantum efficiencies of the hybrid materials?

Reply

We measured the photoluminescence quantum yield (PQY) and added the data with the corresponding discussion.

  1. It will be interesting to check the thermal stability of the hybrid materials.

Reply

We have conducted the additional experiments of in-situ PL measurements of BoronLightSil at different temperatures. It was established that up to 90 C the hydrophobic BoronLightSil still demonstrated 25 % PL intensity and preserved its structure. It was surprising because a pure SiO2-aerogel usually irreversibly changes its structure at temperature above 80 C.

  1. Comment on the stability of the boron-based materials?

Reply

The stability of synthesized boron-containing luminescent hybrid was proved by the study of cathodoluminescence. For the first time for boron-based coordination compound with 8-hydroxyquinoline we succeeded to detect cathodoluminescence.

  1. Thorough comparison between Alq3 and current material should be provided considering both their synthesis and luminescence properties to stablish the correlation between them.

Reply

We made the additional experiments which let’s confirm that:

“It should be notice that LightSil hybrid based on Alq3 [3] has the PQY 45% (measured in this research), but it decayed in several days under natural environment condition.”

Reviewer 2 Report

This paper deals mainly with the synthesis and luminescent characterization of luminescent hybrid material based on silica aerogel and a boron-containing coordination compound with 8-hydroxyquinoline. Although a fine set of luminescent characterizations was carried out, data discussion lacks structural support regarding the complex formation and how the complex interacts with the silica aerogel. Due to this reason, I suggest the paper to be rejected.

- How do the authors make sure that they have formed the 8-Hq complex with Boron? What is the structure of the complexes? How many ligands are coordinated to boron? No structural data is presented. Authors should use basic techniques for structural and compositional characterization such as X-ray diffraction, FTIR, elemental analysis, etc. Without these characterizations, it is difficult to understand the luminescent behavior of the systems.

- Authors should present a figure with ligand structure, complex structure and final hybrid structure instead of basic schematic reactions.

- Authors claim that a hybrid system was formed, but no data confirm that the complex is covalently bonded to the silica surface, which would characterize the system as a hybrid. The authors need to carry out more analyses (for instance, FTIR) to prove that the system is in fact a hybrid and not a composite.

- This paper is too long. There are too many figures and the way the results are discussed look like a lab report and not a paper. Moreover, the language should be revised.

Author Response

- How do the authors make sure that they have formed the 8-Hq complex with Boron? What is the structure of the complexes? How many ligands are coordinated to boron? No structural data is presented. Authors should use basic techniques for structural and compositional characterization such as X-ray diffraction, FTIR, elemental analysis, etc. Without these characterizations, it is difficult to understand the luminescent behavior of the systems.

Reply

This is the problem with 8-Hq complexes with boron to obtain the crystalline sample in amounts enough for single crystal formation. We found out that B-based compound with 8-Hq can be stabilized by a substrate (boron oxide, SiO2-aerogel) or by a solvent (isopropanol, tetrahydrofuran). At drying, the coordination compound is decayed. The size or thickness of the product could be estimated as several nanometers. This is enough to demonstrate luminescence but it is not enough to establish the compound structure. No FTIR neither XPD could solve this problem. We sublimated a thin layer of boron oxide on a corundum substrate and treated it in *-Hq vapor. We obtained the luminescence, which was the same as at the direct synthesis at the same conditions with boron oxide wires. ICP-MS analysis showed that the dissolved layer of a luminescent compound contained boron in amounts close to that of boron oxide layer. It means that during the synthesis we fully converted boron oxide film in boron luminescent compound.

- Authors should present a figure with ligand structure, complex structure and final hybrid structure instead of basic schematic reactions.

Reply

Agreed. Done.

- Authors claim that a hybrid system was formed, but no data confirm that the complex is covalently bonded to the silica surface, which would characterize the system as a hybrid. The authors need to carry out more analyses (for instance, FTIR) to prove that the system is in fact a hybrid and not a composite.

 Reply

Hybrid materials could be formed not only with covalent bonding but also with van der Waal bonds. Because we did not synthesize an individual substance but hybrid with small amount (<1%) of luminescent constituent it is difficult to find the method to analyze it crystal structure. It is not the same as a synthesis of a new complex compound and crystal growth from solution with further X-ray study of a crystal structure. Indeed, this BLC exists only on a substrate or could be stabilized by solvent molecules in solvent media.

- This paper is too long. There are too many figures and the way the results are discussed look like a lab report and not a paper. Moreover, the language should be revised.

Reply

We deleted some details of determination of physical and structural properties of aerogel-based hybrids, which were described in details in another article. We have also changed some scheme of aerogel synthesis on chemical reaction schemes.

Round 2

Reviewer 2 Report

The authors addressed all points raised by this reviewer and due to that, I suggest the paper to be accepted.